# The Path to New Pediatric Vaccines against Pertussis

**DOI:** 10.3390/vaccines9030228

**Published:** 2021-03-05

**Authors:** Camille Locht

**Affiliations:** Univ. Lille, CNRS, Inserm, CHU Lille, Institut Pasteur de Lille, U1019-UMR 8204-CIIL-Center for Infection and Immunity of Lille, F-59000 Lille, France; camille.locht@pasteur-lille.fr

**Keywords:** whooping cough, acellular vaccines, whole-cell vaccines, cocoon vaccination, maternal immunization, nasal vaccination, live attenuated vaccines

## Abstract

Whooping cough, or pertussis, mostly caused by *Bordetella pertussis*, is a respiratory disease that affects all age groups, but severe and fatal pertussis occurs almost exclusively in young children. The widespread use of whole-cell and, more recently, of acellular vaccines has substantially reduced the disease incidence. However, it has not been eliminated in any part of the world and has made a worrisome rebound in several areas. Cocoon and maternal immunization have been implemented in several countries but have their intrinsic limitations. To effectively control pertussis, novel vaccines are needed that protect against disease and prevent *B. pertussis* infection and transmission, which is not the case for current vaccines. Several approaches are contemplated, including alternative administration routes, such as nasal immunization, improvement of acellular vaccines by adding more antigens and T-cell-promoting adjuvants, and the development of novel vaccines, such as outer membrane vesicles and live attenuated vaccines. Among them, only a live attenuated vaccine has so far been assessed for safety and immunogenicity in preclinical models other than mice and is in clinical development. Before any of these vaccines can be used in neonates, extensive safety and immunogenicity assessment in pre-clinical neonatal models and in carefully designed clinical trials is necessary. The aim of this review is to discuss the current pertussis problem, implemented strategies to resolve it, the value of animal models and novel vaccine approaches.

## 1. Introduction

Although pertussis or whooping cough, mainly caused by the Gram-negative bacterium *Bordetella pertussis*, can affect all age groups, the disease is particularly severe and life-threatening in young infants [1]. However, adults can also occasionally develop severe pertussis symptoms and often develop prolonged cough, although the disease is usually not fatal in adults [2]. Importantly, however, infected adults and adolescents can constitute a significant reservoir for serious disease in young children, and seroepidemiological studies have suggested that oligosymptomatic pertussis may occur from 1 to 5% per year in the adult population [3]. Pertussis is a strictly human disease, and besides humans and great apes, no other natural reservoir of *B. pertussis* is known, which implies that it is theoretically eradicable by vaccination [4].

Before the vaccine era, pertussis was one of the most frequent deadly infectious diseases for children. In the beginning of the 20th century, close to 0.5% of all children born in the United States were expected to die of pertussis during their first year of life [5], and the calculated average pertussis attack rate in this country was 872/100,000 population [6]. The vast majority of cases occurred in children less than 5 years of age, and less than 3% of the cases was observed in persons older than 15 years of age. Death also occurred almost exclusively in infants [7]. Pertussis-linked deaths were rare in individuals over 15. In addition to the high death rates, infant pertussis also caused long-term sequelae, including permanent brain damage and developmental retardation. The disease occurred in cycles of roughly every 3 years, depending on the birth rate, suggesting that *B. pertussis* infection leads to sterilizing immunity in pertussis survivors [8]. Consistently, second attacks occurred seldomly, and, unlike for measles and mumps, adult pertussis was rare. However, adult pertussis might have been overlooked due to its atypical clinical manifestation, and “grandmothers’ whooping cough” was not uncommon, suggesting that infection-induced immunity is not lifelong. Nevertheless, adult pertussis is usually mild, but it can serve as a very effective *B. pertussis* reservoir [9]. The aim of this review is to discuss the effect of pertussis vaccination, the current pertussis problem, already implemented strategies to resolve it, the value of animal models to study pertussis and novel vaccine approaches.

## 2. Effect of Pertussis Vaccination and the Recent Resurgence of Pertussis

The introduction and wide-spread use of whole-cell pertussis vaccines (wP) starting in the middle of the previous century, followed by their replacement with less reactogenic, acellular vaccines (aP) in most high-income countries, have drastically reduced the pertussis incidences and associated mortality [10]. Nevertheless, despite high global vaccination coverage [11] pertussis has not disappeared in any region of the world and has remained endemic in most countries. Sporadic outbreaks continue to occur frequently [10]. The most recent estimates indicated between 7 and 40 million global annual pertussis cases and 38,000 to 670,000 associated deaths in children younger than 5 years [12]. Although this represents a decrease in the numbers of global cases and deaths compared to previous estimates, pertussis has made a spectacular come-back in some countries, especially those that have switched from wP to aP [13]. This rebound is also associated with increased pertussis-linked mortality in infants. The reasons for this resurgence are still being discussed [14,15] and include genetic adaptation of circulating strains to escape vaccine pressure, fast waning of aP-induced immunity and the failure of the current vaccines to prevent *B. pertussis* infection and transmission. This latter reason may in fact be the most parsimonious explanation for the resurgence of pertussis in aP-using countries [16].

Several strategies have been proposed to solve the current pertussis problem. They include neonatal vaccination, cocoon vaccination and maternal vaccination during pregnancy. Pertussis immunization with combined diphtheria-tetanus-pertussis vaccines usually starts at 6 weeks to 3 months after birth and often requires at least three doses given at 1 to 2 months intervals for optimal protection, implying that infants younger than 5 months are either not protected or that protection is sub-optimal. Yet, it is in the 0- to 5-year-old age group where pertussis is most severe and where most pertussis-linked deaths occur [12]. At-birth immunization against pertussis has been tempted already in the 1940s with wP, but was found to result in inadequate responses to *B. pertussis* antigens and in persisting non-responsiveness to subsequent vaccination [17]. Therefore, neonatal vaccination has not been implemented. When aP became available, at-birth immunization was attempted again, followed by the regular three-dose primary vaccination schedule. It was found to be safe but was associated with significantly lower antibody responses up to 18 months, compared to the children who did not receive the additional vaccine dose at birth [18]. However, the immune-paralysis was not observed when the at-birth dose was a stand-alone aP [19], or when genetically detoxified rather than chemically detoxified pertussis vaccines were used [20,21]. However, at-birth pertussis vaccination was associated with significantly lower antibody responses to hepatitis B, *Haemophilus influenzae* b, tetanus and diphtheria antigens, co-delivered with the pertussis vaccine during the regular primary course of vaccination [21].

“Cocoon vaccination” has been recommended in several countries. It consists in vaccinating household contacts of neonates in order to limit the transmission of *B. pertussis* to the new-born, and was based on the fact that household contacts are the major source of infant pertussis [22]. However, cocoon vaccination is very difficult to implement, expensive and not cost-effective. Therefore, cocoon vaccination rates remain generally very low, even in high-income countries [23], and not all vaccinated household contacts mount a robust immune response. Logistic and financial barriers often hinder widespread cocoon vaccination, as it requires government-funded vaccination programs, free-of-charge vaccines and aggressive communication strategies to achieve roughly 75% vaccination coverage [24]. However, even when free pertussis vaccination was offered to all household contacts of a newborn, and cocooning could be successfully implemented, it did not reduce pertussis illness in infants less than 6 months of age [25]. Furthermore, since optimal antibody responses to vaccines take at least two weeks to develop, post-partum cocooning would still leave neonates unprotected during the first weeks of life. Most importantly, however, while protective against pertussis disease, pertussis vaccination has at best limited effects on the prevention of colonization by and transmission of *B. pertussis* [26]. Furthermore, experimental studies in baboons and mice suggest that acellular pertussis vaccination may result in prolonged nasal carriage of *B. pertussis* [26,27,28,29].

The most effective strategy to prevent severe pertussis in young infants is maternal immunization in the second or third trimester of each pregnancy. Many studies have shown that tetanus-diphtheria-acellular pertussis (Tdap) vaccination is safe for both mother and child and highly effective in preventing severe and deadly pertussis in the new-born (for a recent review, see [30]). Protection in the new-born is due to the active transplacental transport of the maternal aP-induced antibodies to the fetus. Maternal aP immunization during pregnancy is thus now recommended in several countries, but has so far met with limited compliance [31]. In the US, for example, pregnancy vaccination coverage reaches barely 55% [32]. Barriers to pregnancy vaccination include the perception of low prevalence and severity of the disease, lack of information about vaccine recommendations, concerns about safety for mother and child, as well as economic status and educational levels [31]. Furthermore, studies have shown that maternal antibodies may interfere with the immune responses to wP in the infants [33,34], although less so with the responses to aP [30]. However, since today most children in the world receive wP for their primary immunization series, the clinical consequences of blunting by maternal aP-induced antibodies need to be carefully evaluated.

## 3. The Importance of Animal Models

Given the high contagiousness of *B. pertussis* in human populations, second to measles only for respiratory pathogens [35], it has become clear that novel vaccines are needed that not only prevent disease, but also infection and transmission [36]. Animal models have been instrumental in deciphering the immune responses to infection by *B. pertussis* and after wP or aP vaccination and have identified both humoral and cellular effector mechanisms of protection [37]. Mouse, rabbit, rat, guinea-pig, puppy, piglet, Taiwan monkey, Rhesus monkey [38] and, most recently, baboon [39] models have been used to study pertussis pathogenesis and immunity, each of which with its inherent advantages and limitations.

Mice are by far the most widely used animals to study pertussis, mostly for cost reasons and because of the availability of a wealth of mouse-specific reagents and gene knock-out animals. Mice can be infected by aerosol or nasal drops and develop leukocytosis upon infection by *B. pertussis*, a hallmark of severe human pertussis. However, mice do not cough and cannot transmit *B. pertussis* to littermates. Nevertheless, mice have been extremely useful to decipher protective immune mechanisms against pertussis and *B. pertussis* infection, and have been instrumental in pertussis vaccine development and potency assessment. In the 1940s Kendrick and Elderling developed a standardized mouse intracranial challenge model to test the potency of commercial wP [40], a model still in use to test the potency of wP vaccine lots. Potency in this model correlated with protective efficacy against pertussis in children [41]. Later, a mouse respiratory challenge model was used to evaluate the potency of aP, and a correlation was found between the rate of *B. pertussis* clearance from the lungs of mice after aerosol exposure and vaccine efficacy in children [42]. In addition, this model also highlighted the roles of humoral and cell-mediated immunity, in particular Th1 type responses, in protection against pertussis.

*B. pertussis* is a strictly mucosal pathogen and disseminates only very rarely out of the respiratory tract. Disseminated disease has only been observed in highly immune-compromised patients [43,44], which can be recapitulated in IFN-γ receptor knock-out mice [45]. The strictly mucosal habitat of *B. pertussis* suggests that mucosal immunity may also play an important role in the control of the infection [46], yet current vaccines are usually injected intra-muscularly and therefore are poor inducers of mucosal immunity. Mouse studies have indeed shown the role of mucosal secretory IgA (SIgA) [47] and of tissue resident CD4^+^ memory T (T_RM_) cells [48], especially those that produce IL-17 [23], as well as of neutrophils in the clearance of *B. pertussis*, particularly from the nasal cavity.

The recently developed baboon model recapitulates the nearly full spectrum of human pertussis, including paroxysmal cough [39] and airborne transmission of *B. pertussis* from animal to animal [49]. aP vaccination of baboons protected these animals from pertussis disease, but did not prevent infection and transmission [26], and in fact, similar to mice [27,28,29], prolonged nasal carriage of *B. pertussis*. In contrast, wP vaccination did not prolong nasal carriage in baboons, which cleared the infection faster than non-vaccinated baboons, although much less efficiently than convalescent baboons [26]. Like humans [50] and mice, baboons immunized with aP also generated a mixed Th2/Th1 response, without a significant Th17 response, while infection with *B. pertussis* induced strong Th17 and Th1 responses in this species [51]. In baboons infected with *B. pertussis* local IL-17 responses in the nose were long-lasting. The baboon model has also been used to examine the effect of maternal immunization and confirmed that maternal vaccination with aP protected the offspring against pertussis disease, but not against infection by *B. pertussis* [52]. Using this model, it was further established that maternal vaccination with pertussis toxoid (PT) is sufficient to provide protection against whooping cough disease [53], which was confirmed by the use of humanized anti-PT monoclonal antibodies in this model [54].

Taken together, the data obtained from work on animal models together with clinical data in humans suggest that serum IgG responses that neutralize PT are critical for protection against severe pertussis disease. However, the threshold levels for protection against disease still remains to be established. It may be possible that the baboon model will provide valuable information about these threshold levels. The combined human and animal studies have also shown that the levels of PT-neutralizing antibodies induced by aP wane rapidly, although perhaps less rapidly than antibodies induced by genetically detoxified pertussis vaccines.

Prevention of infection and transmission requires mucosal immunity, both SIgA in the nasal cavity and IL-17-producing T_RM_ cells in the nasal tissues, as is induced by natural infection with *B. pertussis*. Memory T and B cells in the respiratory tract may also be associated with durable protection, consistent with the fact that infection induces longer lasting immunity than vaccination with wP or aP [55]. Novel vaccines that induce both sufficient PT-neutralizing serum antibodies and nasal SIgA together with IL-17-producing T_RM_ cells would therefore be expected to provide long-lived protection against both pertussis disease and colonization by *B. pertussis* and therefore prevent pertussis spread [56]. However, the threshold of these immune responses required to interrupt circulation of *B. pertussis* is still unknown.

## 4. Novel Pertussis Vaccines

Multiple attempts have been made to improve pertussis vaccines, and several recent review articles have discussed novel approaches for improved pertussis vaccines [57,58,59,60]. Various strategies have been contemplated, including improvements of current aP, new wP and entirely novel formulations, such as outer membrane vesicles (OMV) and live attenuated vaccines. Alternative vaccination routes, such as intranasal and cutaneous immunization have also been explored.

Pertussis vaccines are usually administered intramuscularly. As an alternative, epicutaneous delivery of a stand-alone aP vaccine, consisting only of genetically detoxified PT was tested and found to be safe, but the resulting immune responses were poor [61], although this approach showed promise in a mouse model [62]. Immunogenicity could, however, be improved when the skin was prepared by a laser beam prior to vaccine application [61].

Although the epicutaneous route may potentially induce PT-neutralizing serum antibodies, even in the absence of adjuvants, it is unlikely to induce potent mucosal immunity in the respiratory tract. As an alternative, aerosol vaccination with wP has been tempted in human volunteers in the 1970s. It was found to be safe and resulted in a two-fold increase in anti-*B. pertussis* IgA in the respiratory secretions but did not induce serum antibody responses [63]. Serum antibody responses could be detected when high doses of wP were repeatedly administered nasally, but this still induced poor serum responses to PT [64].

Mucosal vaccination usually results in poor immune responses, unless live vaccines or potent mucosal adjuvants are used. One of the most potent mucosal adjuvants is cholera toxin or its *Escherichia coli* heat-labile toxin homolog, as well as their non-toxic B subunits. Nasal administration of an aP mixed with genetically detoxified heat labile toxin resulted indeed in local and systemic immune responses and accelerated clearance of *B. pertussis* from the lungs of mice [65], but its effect on nasal clearance was not examined. Furthermore, the use of cholera toxin-like adjuvants may be problematic in humans, because of their ability to induce Bell’s palsy [66].

More recently, alternative mucosal adjuvants have been assessed for nasal delivery of pertussis vaccines. Intranasal vaccination with bacterium-like particles used as adjuvant led to IgG and IgA production, reduced lung pathology and bacterial burden in the lungs after intranasal challenge with *B. pertussis* [67]. The effect of this formulation on nasal carriage was not reported. Similarly, intranasal immunization with aP mixed with curdlan, a 1,3 ß-glucan, triggered mucosal immune responses, including *B. pertussis*-specific IgA and IL-17, and it significantly reduced the bacterial burden in the lungs and trachea, and provided a modest tenfold reduction in nasal washes three days after challenge [68]. The induction of resident memory T cells in the respiratory tract could not be demonstrated upon curdlan-adjuvanted vaccine administration.

Unlike curdlan, the TLR2 ligand LP1569 together with the STING agonist c-di-GMP mixed with aP and delivered nasally to mice resulted in the induction of CD4^+^ IL-17-producing T_RM_ cells in the nasal tissue, and also provided a substantial level of protection in both the lung and the nose [69]. Importantly, protection lasted for at least up to 10 months and was correlated with the amount of IL-17^+^CD4^+^CD69^+^ T_RM_ cells in the nasal tissue.

Other TLR ligands have also been tested in mice as adjuvants for aP, such as the TLR9 agonist CpG [70], a strong Th1-promoting adjuvant, the TLR7 agonist SMIP7.10 [71], a strong inducer of both Th1 and Th17 responses, and the TLR4 ligands monophosphoryl lipidA and LpxL2 from *Neisseria meningitidis* [72]. Some TLR agonists have been combined with nanoparticles, such as negatively charged poly(lactide-co-glycolide) nanoparticles, which can be freeze-dried [73,74]. However, these formulations have so far only been tested for their protective effects against lung colonization by *B. pertussis*, and their effect on nasal carriage was not examined.

Other attempts to improve aP consisted in including additional antigens in the current vaccines, based on the observation that protective efficacy of aP increased with increasing numbers of antigens included in the vaccines [75]. Increasing the amount of fimbriae (Fim2 and Fim3) to licensed 5-component aP appeared to reduce lung colonization of mice, even by pertactin-deficient *B. pertussis* strains in a dose-dependent manner, without altering the vaccine-induced biomarkers of reactogenicity [76].

Adenylate cyclase toxin (ACT) is a *B. pertussis* virulence factor important to establish infection and able to down-modulate immune responses to *B. pertussis* [77]. It has been proposed as an additional antigen for aP. Although ACT alone did not appear to result in significant protection against lung colonization by *B. pertussis* in mice, it enhanced aP-mediated protection when added to a low aP dose [78]. It induced neutralizing anti-ACT antibodies and increased the production of anti-PT antibodies, suggesting that the neutralization of the ACT immunosuppressive activity by the elicited antibodies has helped to increase the level of anti-PT antibodies. Similarly, the autotransporter BrkA did not provide lung protection by itself, but enhanced protection by aP composed of PT and filamentous haemagglutinin [79]. Other autotransporters, including SphB1, SphB2, BatB, Vag8, were also assessed as protective antigens against pulmonary colonization by *B. pertussis* [80], but only Vag8 and SphB1 provided some degree of protection. When these two antigens were combined with aP, they appeared to enhance aP-mediated protection in the lungs but not in the trachea.

An attractive novel approach for vaccination against pertussis may be the use of OMV derived from *B. pertussis*, as they contain multiple *B. pertussis* antigens and in-build adjuvant properties. They activate the canonical and non-canonical inflammatory pathways in murine and human macrophages [81] and can be prepared as stable freeze-dried formulations [82]. Importantly, both intraperitoneal and intranasal vaccination with OMVs provided significant protection against lung colonization by *B. pertussis* in mice, at levels similar to those induced by wP control vaccine [83]. Both vaccines stimulated mixed systemic Th1/Th2/Th17 responses in mice upon subcutaneous immunization, but OMV vaccination resulted in milder inflammatory responses than wP vaccination [84], suggesting a better safety profile of OMV over wP. To further improve the safety profile of OMVs, their endotoxicity was decreased by the expression of the *Bordetella bronchiseptica pagL* gene in the *B. pertussis* strain used for OMV preparation [85]. The *pagL* gene codes for a lipid A 3-deacylase, which removes an acyl group of the *B. pertussis* lipo-oligosaccharide (LOS). This resulted in a decreased proinflammatory cytokine induction and neutrophil recruitment, without affecting the protective potential of the OMV against lung colonization by *B. pertussis*. When combined with diphtheria and tetanus toxoids these OMVs offered long-lasting protection in mice for at least up to 9 months [86].

*B. pertussis* can be cultured in two main virulence states, named the *bvg*+ or the *bvg*− mode, in which virulence factors are produced or not produced [87]. In order to provide optimal protection, OMV have to be prepared from *B. pertussis* organisms grown in *bvg*+ conditions [88]. Lack of the virulence factors when the organism is grown in the *bvg*− mode, led to diminished protection, despite the induction of high levels of antibodies and unchanged T helper cell responses.

Intranasal immunization of mice with OMV induced strong mucosal immune responses, with pulmonary and nasal anti-*B. pertussis* IgA, as well as lung resident IgA memory B cells and pulmonary Th1- and Th17-type cytokines [89]. In mice, nasal vaccination, with OMV also resulted in improved clearance of *B. pertussis* in nasal washes over subcutaneous vaccination. Interestingly, CD4^+^ T_RM_ cells producing IFN-γ and IL-17 could be induced in the lungs of mice, even when the OMVs were given intraperitoneally, and expanded after *B. pertussis* infection [90], potentially explaining the long-term protection induced by the OMVs. The induction of nasal T_RM_ cells and protection against nasal colonization were not examined in this study.

## 5. Live Attenuated Pertussis Vaccines

As of today, these novel vaccine candidates have only been tested in mouse models, and no information about their potency in other animal models, such as the baboon challenge model, is yet publicly available. Furthermore, none of them have yet entered clinical development. The only novel pertussis vaccine currently in clinical development is the live attenuated vaccine BPZE1.

The first report on a live attenuated pertussis vaccine was published in 1990 [91]. The attenuation was based on a mutation affecting the biosynthesis of aromatic amino acids, in particular the inactivation of the *aroA* gene by the insertion of a kanamycin-resistance gene. This strain was highly attenuated in a mouse aerosol model, and after repeated aerosol administrations, anti-*B. pertussis* serum antibodies were generated, albeit at low levels, and virulent *B. pertussis* was rapidly cleared from the lungs upon aerosol challenge. The induction of cell-mediated immunity and clearance of *B. pertussis* challenge organisms from the nasal cavity were not assessed in this study. In order to improve immunogenicity of attenuated *B. pertussis* deficient in the biosynthesis of aromatic amino acids, an *aroQ* mutant was constructed [92]. Although not tested side-by-side, the *aroQ* mutant appeared to persist somewhat longer in the lungs of mice than the *aroA* mutant, and a single intranasal dose of 10^9^ colony-forming units (CFU) resulted in antibody and cytokine responses to *B. pertussis* antigens, as well as in protection against pulmonary colonization when challenge with 10^8^ CFU of virulent *B. pertussis*.

An alternative approach to attenuate *B. pertussis* was the genetic targeting of virulence factors, in particular three *B. pertussis* toxins, as exemplified by the BPZE1 strain [93]. In this strain the dermonecrotic toxin gene was deleted, the *B. pertussis ampG* gene was replaced by the *E. coli ampG* gene, leading to the reduction in tracheal cytotoxin to background levels, and the PT gene was altered so that it codes for a genetically detoxified protein. After intranasal administration, BPZE1 persisted in the mouse lungs for almost as long as the virulent parent strain, but did not induce lung inflammation and pathology. In addition to inducing strong serum antibody and T cell responses, as well as nasal SIgA and CD4^+^ IL-17-producing T_RM_ cells in the nasal tissue [47], a single dose of BPZE1 provided strong protection against lung [93] and nasal colonization [47] by virulent *B. pertussis* in mice. It also induced potent innate immunity, which led to rapid protection, within days after nasal vaccination with BPZE1 [94]. BPZE1 was also shown to induce off-target protective effects against heterologous viral [95,96] and bacterial [97] infections, as well as against non-infectious inflammatory diseases (for review see [98]), which most likely are due to its ability to induce potent innate immune responses.

BPZE1 has undergone extensive safety assessment in pre-clinical models. In a lethal challenge model in infant mice, it did not cause weight loss, lung pathology or mortality, in contrast to its virulent parent strain [99]. In IFN-γ receptor knock-out mice it did not disseminate out of the respiratory tract, while aerosol infection with virulent *B. pertussis* resulted in dissemination to the spleen and liver and caused atypical pathology in these mice [99]. In MyD88 knock-out mice, intranasal infection with 10^4^ CFU of virulent *B. pertussis* resulted in strong lung inflammation, exacerbated bacterial growth in the lungs, weight loss and 100% mortality within 10 days after infection. In contrast, MyD88 knock-out mice intranasally infected with an up to 100-fold higher dose of BPZE1 did not experience any weight loss, suffered no lung pathology and survived up to the end of the experiment [94].

Genetic stability of BPZE1 was assessed after 20 and 27 weeks of continuous passaging in vitro and in vivo in mice, respectively. PCR and DNA sequence analyses showed that the strain maintained all original genetic characteristics after these passages [100]. A recently developed lyophilized formulation of the vaccine was also found to be biologically stable after more than 2 years of storage at −20, +4 or +22 °C [101].

In a juvenile baboon model, BPZE1 was found to transiently colonize the nasopharynx, but did not induce any whooping-cough-like symptoms, no elevated white blood cell counts and no physical abnormalities or changes in biological parameters, even with a dose of 10^10^ CFU [102]. A single administration of BPZE1 induced high titers of serum IgG and IgA against *B. pertussis* antigens in baboons, and protected the animals from pertussis disease after infection with a very high dose of a highly pathogenic clinical *B. pertussis* isolate. Importantly, a single administration of BPZE1 also substantially diminished the bacterial load in the naso-pharyngeal washes after challenge with the virulent *B. pertussis* isolate. In the baboons vaccinated with 10^10^ CFU of BPZE1, the average bacterial challenge load was reduced by 99.998% compared to the non-vaccinated controls. Similar results were more recently obtained with a live pertussis vaccine, based on the same attenuation approach as for BPZE1, in a rhesus macaque model [103].

BPZE1 is now in clinical development and has completed two phase 1 studies in adult human volunteers [104,105]. Two phase 2 studies are currently under way [NCT03541499 and NCT03942406]. The phase 1 studies have shown that BPZE1 is safe in the humans, able to transiently colonize the human respiratory tract and to induce antibody responses to *B. pertussis* antigens after a single nasal administration. Both phase 1 studies were carried out in Sweden with volunteers who had never been vaccinated with pertussis vaccines and had not knowingly experienced pertussis disease in the past. However, this does not exclude that they may have had encountered *B. pertussis* in the form of a silent infection prior to the enrolment for the phase 1 studies. Both studies were dose-escalation studies, the first with 10^3^, 10^5^ and 10^7^ CFU of a liquid formulation of BPZE1 given as 100 µl nasal drops, and the second with 10^7^, 10^8^ and 10^8^ CFU of the liquid BPZE1 formulation given as 400 µL nasal drops in each nostril. In the first study, even with the highest dose of 10^7^ CFU only 40% of the vaccinees were colonized and developed antibody responses to *B. pertussis* antigens [104]. Interestingly, those subjects that were not colonized by BPZE1 had high pre-existing levels of antibodies to pertactin, filamentous haemagglutinin and fimbriae, suggesting that they had recently been asymptomatically infected with *B. pertussis*. In the second trial, volunteers with pre-existing antibodies to *B. pertussis* antigens were excluded, and naso-pharyngeal colonization by BPZE1 could be detected in more than 80% of the volunteers [105]. Furthermore, in the highest dose group (10^9^ CFU), all subjects sero-converted. These observations further confirm the notion that prior asymptomatic infection by *B. pertussis* prevents subsequent infection, in this case by the attenuated *B. pertussis* strain BPZE1. Importantly, in addition to a broad antibody response, the 10^9^ CFU dose also induced potent cell-mediated immunity [106]. The current phase 2 trials are carried out with the 10^9^ CFU dose of a stabilized, lyophilized BPZE1 formulation [101] administered as a nasal spray, and is expected to provide proof of concept that nasal vaccination with BPZE1 induces mucosal immunity and prevents subsequent infection in humans [NCT03942406].

## 6. Conclusions

The development of novel vaccines against pertussis is a challenge, and most vaccine candidates have not yet gone beyond testing in murine models of lung colonization. Only few studies have investigated the effect of the novel vaccine against nasal colonization, and only two of them have been tested in non-human primates. One vaccine, BPZE1, is currently in advanced clinical development. Up to now, all these vaccine candidates have only been tested in adult or juvenile models, but not yet in neonates, the most vulnerable population for severe and deadly pertussis. The path to neonatal immunization with novel pertussis vaccines has to overcome two main hurdles: extensive safety evaluation and the demonstration of protective immunogenicity in this particular population. It is unlikely that additional antigens to current aP will improve immunogenicity of these vaccines in the new-born. Whether this can be achieved with novel adjuvants or with OMVs remains still to be investigated. It has been documented that natural infection by *B. pertussis* very early in life (as early as 2 weeks after birth) induces strong Th1-type T cell responses [107], providing hope that live attenuated pertussis vaccines may be able to trigger protective immunity in neonates. That live attenuated vaccines are potent inducers of protective immunity at birth is not unprecedented. As an example, the live attenuated anti-tuberculosis vaccine, the bacille Calmette-Guérin, was shown to be highly protective against severe and disseminated tuberculosis, when given at birth [108], and is currently recommended by the World Health Organization for at-birth vaccination against tuberculosis. Before clinical assessment in neonates, the safety issue of novel pertussis vaccines needs of course to be carefully addressed in pre-clinical models. The recently developed neonatal baboon models [52,53] will undoubtedly be instrumental to evaluate safety and efficacy of novel pertussis vaccines in this age group, and current efforts in that direction are expected to soon provide important information on these aspects. Meanwhile, novel vaccines that have demonstrated protection against *B. pertussis* infection and transmission may be valuable tools for cocoon vaccination in older populations, which, in conjunction with maternal immunization using current vaccines, may help in reducing severe infant pertussis, while waiting for novel vaccines to be applicable to neonates.

## Data Availability

Not applicable.

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
