# Peer review of "The Path to New Pediatric Vaccines against Pertussis"

_vaccines, 2021, doi:10.3390/vaccines9030228_

Round 1

Reviewer 1 Report

The manuscript is very interesting since it addresses an important issue for public health and describes some strategies proposed to solve the current pertussis problem.

I have only a few of suggestion in order to improve the quality of the manuscript and the clarity.

Particularly, I suggest to state the aim of the review both in the abstract and at the end of introduction section.

Furthermore, I think should be interesting for the readers a paragraph specifically describing the epidemiology of pertussis worldwide and in Europe, risk factors for transmission and finally and implemented vaccination programmes and recent data on pertussis vaccination status among pregnant women and infant (compliance). Moreover, a brief discussion on possible barriers to vaccine programmes implementation should be added.  

Author Response

The aim of the review was now added both at the end of the abstract and at the end of the introduction (see lines 16 and 17, and lines 41-43 of the revised version). Although this article is not about the pertussis epidemiology, I have added a short section on the current global epidemiology (see lines 48-52 of the revised version). I also discussed the difficulties and barriers to implement vaccination programs (cocooning and maternal immunization, in lines 76-84, and lines 92-95, respectively, in the revised version, with the corresponding references 23-25 and 32, respectively).  

Reviewer 2 Report

I was invited to review the paper entitled "The Path to New Pediatric Vaccines Against Pertussis". It is a review on the history of pertussis vaccination and on the challenge of  novel pertussis vaccination.

The paper is easy to read and it deeply describe the pertussis vaccination strategy from the past to recent years. I want to congratulate with the Author for the excellent work, focusing across all aspects of this vaccination and his related strategies.

In my opionion, the Author missed just one point: the burden of adult pertussis. Frequently adults do not boost the DTP vaccination across their life, allowing the circulation of the bacteria. In addition, it can involve health care workers, exposing also patients to this disease.

Author Response

The burden of adult pertussis is now briefly discussed in lines 22-27 of the revised version, with the corresponding references (2 and 3 in the revised version).